# Deaminase-Independent Mode of Antiretroviral Action in Human and Mouse APOBEC3 Proteins

**DOI:** 10.3390/microorganisms8121976

**Published:** 2020-12-12

**Authors:** Yoshiyuki Hakata, Masaaki Miyazawa

**Affiliations:** 1Department of Immunology, Kindai University Faculty of Medicine, 377-2 Ohno-Higashi, Osaka-Sayama, Osaka 589-8511, Japan; masaaki@med.kindai.ac.jp; 2Kindai University Anti-Aging Center, 3-4-1 Kowakae, Higashiosaka, Osaka 577-8502, Japan

**Keywords:** APOBEC3, deaminase-independent antiretroviral function, innate immunity

## Abstract

Apolipoprotein B mRNA editing enzyme, catalytic polypeptide-like 3 (APOBEC3) proteins (APOBEC3s) are deaminases that convert cytosines to uracils predominantly on a single-stranded DNA, and function as intrinsic restriction factors in the innate immune system to suppress replication of viruses (including retroviruses) and movement of retrotransposons. Enzymatic activity is supposed to be essential for the APOBEC3 antiviral function. However, it is not the only way that APOBEC3s exert their biological function. Since the discovery of human APOBEC3G as a restriction factor for HIV-1, the deaminase-independent mode of action has been observed. At present, it is apparent that both the deaminase-dependent and -independent pathways are tightly involved not only in combating viruses but also in human tumorigenesis. Although the deaminase-dependent pathway has been extensively characterized so far, understanding of the deaminase-independent pathway remains immature. Here, we review existing knowledge regarding the deaminase-independent antiretroviral functions of APOBEC3s and their molecular mechanisms. We also discuss the possible unidentified molecular mechanism for the deaminase-independent antiretroviral function mediated by mouse APOBEC3.

## 1. Introduction

Humans and mice have at least 11 and 5 proteins, respectively, which belong to the activation-induced cytidine deaminase and apolipoprotein B mRNA editing enzyme, catalytic polypeptide-like (AID/APOBEC) family. These include AID, APOBEC1, APOBEC2, APOBEC3A, 3B, 3C, 3D, 3F, 3G, 3H, and APOBEC4 for humans and AID, APOBEC1, APOBEC2, APOBEC3, and APOBEC4 for mice [1]. Most AID/APOBEC family members act as mutator enzymes for DNA or RNA, and perform essential roles in host defense mechanisms such as innate and adaptive immunity to combat invading microorganisms [2]. Mutations induced by the AID/APOBEC family on viral genomes in the innate immune stage restrict the viral replication, and thus its pathogenesis, whereas those on host genome DNAs are involved; for example, in the affinity maturation of B cell antigen receptor, which occurs in the adaptive immune stage [3,4]. Among them, APOBEC3 proteins (APOBEC3s) are intrinsic restriction factors that protect the host from both threats by infection from exogenous viruses and loss of genome integrity by retrotransposons [5]. They are also known to be associated with human tumorigenesis and metastasis [6,7,8]. One of the molecular mechanisms mediating the functions of APOBEC3s relies on its deaminase activity by which cytosines on viral and host genomic single-stranded DNAs are converted to uracils, resulting in C-to-U mutations. For example, human APOBEC3G generates several C-to-U mutations on the HIV-1 minus-strand DNA during the reverse transcription process, resulting in the plus strand of viral DNA with G-to-A hypermutation that can be catastrophic for HIV-1 [9,10,11]. Characteristic mutations, called kataegis, are detected in the genomes of cancer cells, and the observed base substitution patterns are consistent with those mediated by APOBEC3A and/or APOBEC3B [6,8,12,13]. However, it is now evident that the antiviral function and promotion of tumor development by APOBEC3s are not mediated solely via their deaminase activities. The deaminase-independent pathway, in addition to the enzymatic activity, is important for APOBEC3s to exert the full extent of their biological functions. For instance, an APOBEC3G mutant deficient of deaminase activity still inhibits HIV-1 reverse transcription [14,15,16]. There seem to be several steps in the reverse transcription that are targeted by APOBEC3s in a deaminase-independent manner. Furthermore, in addition to reverse transcription, other retroviral replication processes are targeted by a deaminase-independent antiretroviral function of APOBEC3s. Recently, we found that mouse APOBEC3 (mA3) without enzymatic activity dysregulates autoprocessing of murine leukemia virus (MuLV) Pr180gag-pol precursor proteins within virions, resulting in the reduction of Pr65gag processing by viral protease (PR) followed by the maturation of the viral core [17]. We also observed a reduction in MuLV virion production from cells expressing either wild-type or catalytically inactive mA3, implying that there seems to be an unknown deaminase-independent molecular mechanism that interferes with MuLV replication.

In this review, following a brief introduction to the AID/APOBEC family members and the deaminase-dependent APOBEC3 function, we focus on the deaminase-independent antiretroviral functions of APOBEC3s. We also discuss one more possible molecular mechanism for the deaminase-independent pathway that mA3 seems to exert on MuLV replication.

## 2. AID/APOBEC Family

AID/APOBEC family members are evolutionarily conserved cytidine deaminases that act on the preferred nucleotide contexts of DNA and RNA [1]. They play roles mainly in innate and adaptive immune systems [5]. In addition to editing nucleotides in pathogen genomes, some of these enzymes target host genomes. In fact, a signature attributed to AID/APOBEC enzymatic activity has been observed in the genomes of human cancers [6,8,12,13].

Of the 11 members of the human AID/APOBEC family, AID is thought to be the oldest, and was firstly identified as an essential factor for the function of activated B cells in the germinal center [18]. AID acts on immunoglobulin (Ig) genes in activated B cells to initiate both somatic hypermutation and class switch recombination [19]. In these biological processes, AID induces a significant level of cytosine deamination on Ig genes. The variable regions within Ig heavy and light chain genes are edited by this enzymatic activity to produce a diverse set of Ig repertoires in which only Igs with higher affinity to the stimulating antigen are selected in the next step, resulting in the affinity maturation. Similarly, cytosine residues in switch regions of the Ig heavy chain gene locus are also edited to mediate class switch recombination. In contrast to these physiologically important functions, deregulated expression of AID under the activation of intracellular signaling induced by microbial pathogen infection can lead to cancer development [20,21,22]. These results suggest that AID is involved in tumorigenesis, possibly through its enzymatic activity. Furthermore, AID can also edit viral genomes, such as that of hepatitis B virus (HBV), to compromise viral replication [23,24,25]. However, AID expression has been reported to be stimulated by certain cytokines that appear to inhibit HBV via a deaminase-independent mechanism [26,27].

APOBEC1 was the first member of the AID/APOBEC family to be identified and acts in lipid metabolism via its RNA editing activity. Editing a cytosine within the apolipoprotein B pre-mRNA encoding apolipoprotein B100 by this enzyme generates a new stop codon in the mRNA, resulting in the translation of a shorter form named apolipoprotein B48 [28,29]. Several other cellular mRNAs are possibly targeted by APOBEC1 [30,31,32,33]. In addition to RNA editing activity, APOBEC1 is capable of mutating DNA and inhibiting some types of viruses and retroelements [34,35,36,37,38,39,40,41,42,43,44,45,46]. Similarly to AID, ectopic expression of rabbit APOBEC1 causes tumorigenesis in transgenic mice [47].

The physiological function of APOBEC2 is proposed to be related to muscle development in mammals since it is expressed in skeletal and cardiac muscle tissues [48]. However, the significance of its function in life remains unknown, as APOBEC2-deficient mice do not show any significant sign with an abnormal phenotype [49]. Interestingly, APOBEC2 does not appear to have a cytidine deaminase activity even in a bacterial mutation assay, a well-known assay for the evaluation of deaminase activity [39,50]. However, it remains unclear whether APOBEC2 works solely in a deaminase-independent manner or has a rare substrate with an unidentified cofactor.

Like APOBEC2, APOBEC4 does not exhibit deaminase activity, although it has a conserved deaminase domain with some differences from that of other AID/APOBEC family members [50,51]. APOBEC4 is expressed in human and mouse testes [52,53]. Interestingly, APOBEC4 promotes HIV-1 LTR promoter activity and enhances virus production [53]. Recently, chicken APOBEC4 was found to be constitutively expressed in several tissues including bursa of Fabricius. Its expression in several chicken cell lines and primary cells was induced by infection with Newcastle disease virus (NDV) [54]. The report showed that chicken APOBEC4 reduces viral RNA and protein in NDV-infected cells, and acts as an antiviral factor for NDV.

Humans have at least seven *APOBEC3* genes on chromosome 22 in tandem arrangement, APOBEC3A (A3A), APOBEC3B (A3B), APOBEC3C (A3C), APOBEC3D (A3D), APOBEC3F (A3F), APOBEC3G (A3G), and APOBEC3H (A3H). The number of *APOBEC3* genes varies among mammals: horses, cows, and rodents (specifically mice and rats), for example, have six, three, and one per haploid genome, respectively. Differences in the number of *APOBEC3* genes in mammals as well as in the amino acid sequences of APOBEC3s are likely due to positive selection in APOBEC3 molecular evolution, which is attributed to the long-lasting host-virus arms races [55,56,57,58,59,60,61,62]. Bats are thought to be natural hosts of several emerging viruses including Ebola and Marburg viruses which are highly pathogenic to humans. Interestingly, pteropid bats possess 18 putative APOBEC3 coding domains in the haploid genome, the highest variety of APOBEC3 paralogs known to date, probably due to ongoing natural selective pressures posed by pathogens [63]. Thirteen of them are transcriptionally active, and some are indeed catalytically active. Recently, it was reported that the diversity of the *APOBEC3* gene has expanded in simian primates beyond the well-known APOBEC3 locus (chromosome 22 for humans being one example) by a molecular mechanism called retrocopying [64]. In retrocopying, reverse transcriptase (RT) and integrase (IN) of retroelements such as endogenous retroviruses (ERVs) or long interspersed element-1 (L1) could occasionally act on host mRNAs such as APOBEC3 mRNAs, leading to the generation of a new gene copy without introns at a separate location of the chromosome.

Although all APOBEC3 members convert cytosines to uracils in a single-stranded DNA, the preferred target sequences are different to varying degrees among members [40,65,66,67,68,69,70]. For example, human APOBEC3G appears to prefer the CCC (where C is the target base) which is hardly targeted by AID that favors the WRC sequence (W = A/T, R = A/G) [71]. Other human APOBEC3s mainly prefer the TC element, and among these APOBEC3s, the more preferred sequence seems to be TC depending on bases flanking the target C and DNA structures such as open or stem structures. As for functions, APOBEC3s exert antiviral activity against not only retroviruses including HIV-1 and endogenous retroelements but also HBV, adeno-associated virus, foamy virus, Epstein–Barr virus (EBV), coronavirus, and some other viral species [9,10,11,72,73,74,75,76,77,78,79,80,81,82,83,84,85]. APOBEC3s restrict viral replication via deaminase-dependent and/or -independent mechanisms.

## 3. Deaminase-Dependent Functions of APOBEC3s

Among APOBEC3s, human APOBEC3G was the first identified restriction factor against HIV-1 infection. HIV-1, however, antagonizes it with viral infectivity factor (Vif), a key regulator targeting APOBEC3G for proteasome-mediated degradation [74]. APOBEC3G in infected cells is encapsidated into progeny HIV-1 virions through interaction with viral nucleocapsid (NC) protein and RNA if HIV-1 does not express Vif [86,87,88]. Although the number of APOBEC3 molecules that reside in each viral particle remains largely unknown (for most viruses), a previous report estimated that approximately 7 (±4) APOBEC3G can be incorporated into an virion of HIV-1 lacking Vif (HIV-1 (ΔVif)) that produced from human peripheral blood mononuclear cells [89]. Following infection of target cells with APOBEC3G-containing HIV-1 virions, APOBEC3G is released into the cytosol together with the viral genomic RNAs and proteins associating with the RNAs, and catalyzes C-to-U conversions on the minus strand of viral DNA during reverse transcription [90]. These mutations compromise coding sequences for viral proteins and/or produce undesired stop codons, leading to the inhibition of viral replication. When HIV-1 expresses Vif in virus-producing cells, it assembles with APOBEC3G, core binding factor subunit β (CBFβ), and the cullin-RING ubiquitin ligase complex (consisting of cullin 5, ring box protein 2, and elongin B/C) for APOBEC3G degradation [91,92]. In this assembly, CBFβ stabilizes HIV-1 Vif in cells and promotes the degradation of APOBEC3G [91,92,93]. In addition to HIV-1, another primate lentivirus (PLV) Vif such as SIV Vif also hijacks CBFβ for its host APOBEC3s degradation and efficient viral infectivity, although non-primate lentivirus Vif seems not to utilize it [91,93,94]. Primate APOBEC3Gs show their own species-specificity on restriction of PLV infection, and they appear to counteract viruses that are transmitted across species, likely depending on their nature to resist to the Vif function [95,96]. In addition to APOBEC3G, human APOBEC3C, 3D, 3F, and the products of certain 3H alleles are also sensitive to Vif-mediated degradation in HIV-1-infected cells. They work as restriction factors in an essentially similar manner to APOBEC3G [97,98,99]. Interestingly, the amount of RT and its enzymatic activity rate to generate the double-stranded viral DNA influence the extent of the APOBEC3-mediated G-to-A hypermutation [100,101]. It was also reported that APOBEC3B significantly deaminates EBV genomic DNA during viral replication and reduces its infectivity, when the virus has no BORF2, which inhibits APOBEC3B enzymatic activity [85].

All APOBEC3s contain one or two cytidine deaminase domains (CDs), which are also called zinc-coordinating domains (Z-domains), that consist of H-X-E-X_25–31_-C-X_2–4_-C, where X represents any amino acid (Figure 1) [59,62,102]. The Z-domains can be further classified into three types, Z1, Z2, or Z3 based on characteristic amino acids in these domains [62]. As for human APOBEC3s, APOBEC3B, 3D, 3F, and 3G have two Z-domains while APOBEC3A, 3C, and 3H have one domain [62]. In the case of the double domain type of human APOBEC3s, only the C-terminal domain (CTD) is catalytically active [103,104]. In the active catalytic center, a zinc ion (Zn^2+^) is coordinated by the histidine and two cytidines in the Z-domain. A hydroxyl group (–OH) derived from a water molecule is also coordinated to Zn^2+^ as the fourth ligand, and it performs a nucleophilic attack on the C4 site of the cytosine ring in the initial stage of the process of deamination reaction [105]. During the reaction, glutamate in the Z-domain shuttles the proton. The amino group on the cytosine is substituted for an oxygen atom derived from the –OH in the later stage of the reaction, producing a C-to-U substitution and an ammonium ion. In contrast to the CTD, the N-terminal domain (NTD) of human APOBEC3s with double Z-domains is catalytically inactive. However, the NTD plays an indispensable role in the interaction with cellular and viral RNAs, which enables the incorporation of APOBEC3s into retroviral progeny particles [103,104]. Interestingly, the orientation of mA3 CDs is fully reversed compared to those of human APOBEC3G, B, D, and F [106]. The NTD of mA3 is enzymatically active and introduces G-to-A hypermutation in the HIV-1 DNA when mA3 is expressed in cells together with the HIV-1 molecular clone in vitro. On the other hand, the CTD of mA3 is catalytically inactive and works similarly to the NTD of human APOBEC3G. In contrast to APOBEC3s with double CDs, the Z-domain of single-domain APOBEC3s, such as APOBEC3C and APOBEC3H, should perform both functions, that is, nucleic acid binding and deamination [107,108,109,110,111].

When APOBEC3s are aberrantly expressed at the wrong time and/or in excess amount in human cells, its deaminase activity can generate mutations in host genomic DNAs irrespective of pathogen infection. Previously, more than 20 mutational signatures were extracted by analyzing somatic mutation patterns in genomes derived from several human cancers. The one among these mutational signatures could be due to the deaminase activity of APOBEC family members, which probably act in collaboration with DNA replication and/or DNA repair systems [112]. In particular, APOBEC3A and APOBEC3B have received a lot of attention as they localize in the cell nucleus at a steady state, which enables them to directly contact chromosomes [6,8]. Additionally, sequences observed at the APOBEC mutational signature seem to correspond to their preferred target sequences [12,13,112,113,114]. A correlation between APOBEC3B expression level and the number of mutations in genomes was revealed using cancer cells [6,13,114,115]. APOBEC3A was reported to be a major contributor to the APOBEC mutation signature in several breast cancer cell lines and have a significant role in mutagenesis in primary breast cells [116]. While it is not completely understood whether APOBEC3A or APOBEC3B induces mutations at key cytosine residues within certain driver genes leading to tumorigenesis, more recently, it was shown that APOBEC3A causes mutations and drives tumorigenesis [117]. It also remains unclear which tissues in a body and which type of cells in tissues APOBEC3s are induced in response to physiological stimuli such as cytokines and hormones. Recently, RNA editing by some APOBEC3s on mRNAs derived from breast cancer tumors was reported in positive correlation with higher immune responses and survival rates, indicating that APOBEC3 association with RNA and its editing activity could also have a crucial role in tumor development and selection [118].

## 4. Viruses Inhibited by a Deaminase-Independent Function of APOBEC3s

As mentioned above, it is clear that human APOBEC3G inhibits HIV-1 (ΔVif) through a deaminase-dependent mechanism. However, it has also been reported that it represses HIV-1 replication in a deaminase-independent manner [16,119,120,121,122,123,124,125,126,127]. In primary CD4^+^ T cells that express physiological amounts of APOBEC3s, most likely APOBEC3G restricts HIV-1 replication via inhibition of even the initial step of reverse transcription in addition to massive cytosine deamination on viral DNA [128]. A catalytically inactive APOBEC3F still shows an anti-HIV-1 effect, suggesting APOBEC3F also possesses a deaminase-independent antiviral function [129]. HIV-1 reverse transcription is known to be inhibited by a catalytically inactive APOBEC3F mutant [130]. APOBEC3H was also reported to be capable of inhibiting HIV-1 replication without causing significant levels of G-to-A mutations [131]. Furthermore, it inhibits reverse transcription independently of the deaminase activity possibly through preferential binding to the structured RNA element in the HIV-1 genome near the primer-binding site [107,132].

Human T cell leukemia virus type 1 (HTLV-1) is sensitive to the deaminase-independent antiviral function of APOBEC3G [76]. Human APOBEC3G is encapsidated into HTLV-1 virions and restricts its replication without causing significant G-to-A hypermutation [76,104]. Although HTLV-1 does not have an accessory protein like Vif of HIV-1, the C-terminal region of its NC is capable of inhibiting APOBEC3G encapsidation into virions [133]. G-to-A mutations seem to be extensively rare in HTLV-1-positive individuals [77]. A deaminase-deficient APOBEC3G mutant retains the full antiviral activity of APOBEC3G against HTLV-1 infectivity [76]. Furthermore, although deaminase-deficient APOBEC3A and APOBEC3B mutants showed reduced anti-HTLV-1 activity, the corresponding mutant of APOBEC3H haplotype II product retained it. This suggests that APOBEC3H haplotype II restricts HTLV-1 in a deaminase-independent manner [134]. However, a recent report showed that the G-to-A mutations are detectable as a major type of mutation pattern among all detected types of nucleotide changes in HTLV-1 genome sequences of some HTLV-1-carriers, in whom anti-HTLV-1 antibody was hardly confirmed by western blotting screening [135]. In summary, it is feasible that a deaminase-independent antiviral function plays a specified role in the control of HTLV-1 replication, while a deamination-dependent mechanism is also associated with the in vivo restriction of HTLV-1 replication.

HBV is also sensitive to the antiviral function of some human APOBEC3s [80,136,137,138,139]. Before the identification of APOEBC3G as a restriction factor for viruses, the G-to-A hypermutation had already been detected in HBV genomes derived from chronic virus carriers, like that in HIV-1 [140,141]. The G-to-A mutations generated by APOBEC3s were also detected in the HBV genome by differential DNA denaturation-PCR (3D-PCR) [138]. These reports indicate that the deaminase activity of APOBEC3s undoubtedly affects the HBV replication cycle. However, a previous report also indicated that HBV replication is compromised by a deaminase-independent pathway including the inhibition of reverse transcription [142].

The mammalian genome comprises a large number of transposons such as retroelements. The retroelements include endogenous retroviruses (ERVs), which are long terminal repeat (LTR)-positive elements, and L1, which is the most common among non-LTR elements [143]. L1 is the only active autonomous mobile element detected in humans. It is transcribed, and its mRNA is exported into the cytoplasm and translated to ORF1p and ORF2p proteins [144]. These proteins form the ribonucleoprotein complex with L1 mRNA (referred to as L1 RNP complex). This returns to the nucleus, and a new L1 copy is generated in the genome via target site-primed reverse transcription (TPRT). Among over 500,000 L1 copies in the human genome, less than 100 are thought to be active with hot L1 copies that are extensively transcribed [145]. Although transcription of L1 copies is generally suppressed through several mechanisms including CpG methylation in somatic cells [146], they are abnormally expressed in many types of cancer cell, suggesting possible involvement in the initiation and/or progression of cancer [147]. APOBEC3s have been reported to inhibit L1 retrotransposition both in deaminase-dependent and -independent ways [148,149,150,151,152,153]. For example, overexpressed APOBEC3C inhibits L1 reverse transcription in a deaminase-independent manner probably through the interaction with L1 RNP complex [154].

Of note, APOBEC3G appears to be able to interfere with replication of some RNA viruses including measles virus (MV), a negative-strand RNA virus that belongs to *Paramyxoviridae*, through a deaminase-independent mechanism [155]. MV RNA was associated with APOBEC3G and was significantly decreased in the presence of APOBEC3G. However, it remains unknown how the binding of APOBEC3G causes the reduction of MV RNAs.

Severe acute respiratory syndrome coronavirus 2 (SARS-CoV-2) emerged in 2019 and has already spread all over the world [156,157,158,159]. It belongs to the family *Coronaviridae* together with SARS-CoV, MERS-CoV, HCov-OC43, HCoV-229E, HCoV-NL63, and HCoV-HKU1, all of which are pathogenic to humans with severe syndromes or general common colds. Human APOBEC3C, 3F, and 3H inhibited replication of HCoV-NL63 without heavy hypermutation [82]. Both deaminase-dependent and -independent mechanisms were suggested to affect the replication of HCoV-NL63 since the deaminase-dead APOBEC3s still inhibit virus replication with reduced activity compared to catalytically active ones. The GC contents (% of G and C among all nucleotides consisting of a viral genome) of coronaviruses is relatively low; for example, SARS-CoV-2, SARS-CoV, and HCoV-HKU1 have 38%, 41%, and 32%, respectively. Genomic analysis of SARS-CoV-2 revealed that C-to-U changes are generated throughout the viral genome during virus expansion as a major pattern of nucleotide substitutions [160]. However, it remains unknown whether APOBEC3s generate these C-to-U substitutions on viral RNAs. It also remains unclear whether these C-to-U changes lead to a reduction in coronavirus replication and pathogenesis.

The parvovirus adeno-associated virus requires a helper virus such as adenovirus for efficient viral replication in the cell nucleus [161]. This viral genome consists of single-stranded DNA, and its replication is inhibited by APOBEC3A and a deaminase-deficient APOBEC3A mutant [162].

Replication of mouse gammaretrovirus, Friend or moloney MuLV, and betaretrovirus, mouse mammary tumor virus (MMTV), is inhibited by mA3 with few or no G-to-A mutations [75,79,163,164,165,166]. Transgenic mice expressing deaminase-deficient mA3 mutant without endogenous wild-type mA3 showed severe restriction activity for MuLV infection in vivo [167].

As described above, the deaminase-independent antiviral function of APOBEC3s is widely involved in the restriction of many types of virus, strongly suggesting that this function is as necessary as the deaminase-dependent function for the complete host defense system.

## 5. Mechanisms of the Deaminase-Independent Antiretroviral Function of APOBEC3s

Several molecular mechanisms for the deaminase-independent antiretroviral function of APOBEC3s have been reported (Figure 2).

One of the known deaminase-independent antiretroviral mechanisms exerted by APOBEC3s is the inhibition of reverse transcription process in which, for instance, APOBEC3G inhibits the negative-sense (-), single-stranded DNA synthesis [15,168]. Additionally, APOBEC3G oligomers interact with a template viral RNA or (-) ssDNA and subsequently inhibit RT movement, known as a roadblock model [14,169]. Alternatively, it is also proposed that APOBEC3G inhibits this process through direct binding to HIV-1 RT but not through its binding to a template viral RNA [170,171]. A noncatalytic human APOBEC3G mutant, A3GC291S, blocks the accumulation of the HIV-1 early and late reverse transcription products and the proviral DNAs integrated into the host genome. The extent of reduction in proviral copy numbers is higher than that seen for the late reverse transcription products, suggesting that APOBEC3G inhibits proviral DNA integration beyond the reverse transcription process in a deaminase-independent mechanism [122]. The authors also indicated that APOBEC3G interacts with HIV-1 NC and IN in virions, both of which are involved in reverse transcription and integration processes, and that the integration step could be targeted possibly through an interaction of APOBEC3G with the C-terminal domain of IN. Similar to APOBEC3G, APOBEC3F also binds IN, and reduces the accumulation of HIV-1 proviral DNA [122]. Furthermore, APOBEC3F targets the HIV-1 integration step by reducing 3′ processing of viral DNA ends, which is independent of deaminase activity [172]. APOBEC3F is reported to depend more on a deaminase-independent antiviral activity to inhibit HIV-1 reverse transcription than APOBEC3G [124]. APOBEC3G was also reported to reduce tRNA annealing with virus genomic RNA, which is required for the initiation of HIV-1 reverse transcription and subsequent priming for reverse transcription through interaction with NCp7, resulting in inefficient reverse transcription [120,121]. Furthermore, APOBEC3G and APOBEC3F affect the efficiency of the template switching during HIV-1 reverse transcription in a deamination-independent manner, which can result in the accumulation of detrimental insertion and deletion mutations on viral DNA [173].

We and others previously showed that mA3 restricted MuLV in their natural hosts without inducing significant levels of G-to-A hypermutation [79,163,164]. MMTV replication is also reported to be suppressed during reverse transcription by mA3 [165]. These reports suggest that mA3 inhibits murine retroviruses mainly through a deaminase-independent pathway. Coimmunoprecipitation assays revealed that mA3 and its catalytically inactive mutant bind MuLV RT without depending on RNA, and this direct interaction is thought to cause the inhibition of reverse transcription by mA3 (Figure 2) [167]. Recently, we evaluated the effect of mA3 on the activity of MuLV protease (PR) embedded within Pr180gag-pol precursor polyprotein whose autoprocessing generates mature RT and IN in addition to PR, and found that physiologically relevant amount of mA3 interferes with the production of mature PR from the precursor via a deaminase-independent mechanism (Figure 3) [17]. This appeared to be consistent with the previous observation that human APOBEC3G and mA3 interfere with normal Pr65gag processing with an unknown mechanism [174]. There had been no reports showing the direct interaction of APOBEC3s with mature retroviral PR. We demonstrated that the mA3 binding site lies in the region downstream of the amino acid residue 65 of mature PR. GST-pull down and immunoprecipitation assays revealed that human APOBEC3G directly binds both mature RT and IN proteins [175]. Thus, APOBEC3s may be able to associate with all known retroviral enzymes indispensable for viral replication. This may imply that APOBEC3s have several internal domains for contacting with each viral enzyme to control retroviral enzyme activities independently of its deaminase function. We also reported that mA3 binds Pr180gag-pol precursor, as well as mature PR, and this interaction is mediated mainly through the C-terminal half region of mA3, which is the region required for RNA binding when mA3 is encapsidated into viral particles. In contrast, the catalytically active N-terminal half region has only a minor effect on binding to and inhibiting the autoprocessing of Pr180gag-pol. Not only the embedded PR but also the RT and IN sequences are proposed to be involved in Pr180gag-pol autoprocessing [176,177,178,179]. Although we could not determine a region in the precursor responsible for abnormal autoprocessing caused by mA3 in a deaminase-independent manner, we assume that one of PR, RT, and IN regions embedded within the precursor or combination of these regions may be involved in this process. Autoprocessing requires the homodimerization of Pr180gag-pol precursors to activate the embedded PR. Thus, it might be plausible that mA3 may inhibit proper homodimer formation in immature viral particles by binding to the precursor. Alternatively, the mA3 binding may induce structural changes in the precursors, leading to abnormal autoprocessing such as digestion of unusual sites within the dimerized precursors, so that an intact mature PR is not effectively generated. As the Pr180gag-pol autoprocessing is an indispensable step for the production of all viral enzymes, its dysregulation by mA3 in a deaminase-independent manner may profoundly affect viral replication compared to single-step inhibition at reverse transcription or integration.

Recently, it has been shown that the P50 protein, which is translated from an alternative spliced MuLV gag RNA, binds mA3 in cells and inhibits its incorporation into viral particles [180]. Since not all mA3 might be bound by P50 in cells, it is possible that mA3, which is not captured by P50, is incorporated into viral particles and interferes with the autoprocessing of Pr180gag-pol. The autoprocessing dysregulation by mA3 results in inhibition of Pr65gag processing and then of viral core formation. MuLV glycosylated Gag (glyco-Gag) stabilizes viral cores and protects reverse transcription complex from attack by mA3 [181]. The inhibitory effect of mA3 on autoprocessing precedes the glyco-Gag function in the virus life cycle.

The critical question here is to what extent does each deamination-independent antiretroviral activity play a role in overall APOBEC3-mediated antiretroviral activity including the deaminase-dependent pathway. This would depend on a type of virus, APOBEC3s, and cells infected with pathogens.

## 6. A Deaminase-Independent Cellular Function of APOBEC3B

A deaminase-independent function of APOBEC3B influences tumor development [182,183]. For example, APOBEC3B interferes with the tri-methylation at the 27th lysine residue of the histone H3 protein (H3K27me3) epigenetic modification of chromatins, leading to aberrant chemokine expression and subsequent formation of a microenvironment suitable for cancer progression, which seems to be functioning in a deaminase-independent manner [182]. The authors showed that mA3 could also promote the cancer-associated microenvironment formation as does APOBEC3B. Analyses with respect to the involvement of deaminase-independent functions of APOBEC3s in cancer development have just begun. Thus, a deeper understanding of the molecular mechanisms of these deaminase-independent functions will be essential in the area of infectious diseases and cancer research. Such efforts may enable the discovery of novel therapeutic methods for patients with infectious diseases or cancer.

## 7. An Alternative Deaminase-Independent Anti-MuLV Mechanism of mA3

We predict there being at least one more deaminase-independent mechanism by which mA3 inhibits MuLV replication. Previously, we observed that mA3 reduces MuLV viral particle production in the supernatant of cells infected with MuLV or transfected with an MuLV molecular clone despite comparable amounts of viral proteins generated in these cells and the mA3-negative control cells [17]. Martin et al. previously reported that APOBEC3G-containing dot-like structures, which were designated as APOBEC3G complexes, are detected in the cytoplasm, and the formation of APOBEC3G complexes in virus producer cells correlates with a decrease in HIV-1 production [184]. Further, the APOBEC3G complexes reduce the half-life of HIV-1 Gag protein. The reduction of MuLV production by mA3 that we observed may, at least partially, be caused by a similar mechanism via possible mA3 complex formation in the cytoplasm. However, we did not observe a significant reduction in MuLV Pr65gag in the presence of mA3 in virus-producer cells. Furthermore, we found that the reduction of MuLV production is likely mediated by a deaminase-independent action of mA3, as a deaminase-deficient mA3 E73A mutant was able to significantly reduce MuLV production [17]. Thus, our results imply that the viral assembly or later steps of MuLV replication may be impaired by mA3 without enzymatic activity. Of the possible viral replication processes, we assume that the assembly process can be targeted by mA3 as it can associate with viral genomic RNAs (gRNAs) that are incorporated into viral particles, and gRNA is a core material for efficient virus assembly followed by the production of viral particles. York et al. utilized CLIP–Seq analyses to determine the specific regions within HIV-1 RNA that are bound by human APOBEC3s [185]. They showed that APOBEC3s preferentially bind G-rich and/or A-rich regions, and the RNA binding specificity partially mimics that of HIV-1 Gag protein. They also found that the viral RNA binding profile of APOBEC3s did not show a statistically significant correlation with that of HIV Gag in cells, although a similar binding profile was also observed in some regions of viral RNA. We speculate that molecular competition between APOBEC3s and Gag on viral RNA may occur in regions where these two proteins commonly bind, resulting in an inefficient viral assembly. Alternatively, APOBEC3 binding at other specific sites within a viral RNA may inhibit the completion of viral assembly. It was reported that APOBEC3G and HIV-1 NC do not interfere with each RNA binding property and form a ribonucleoprotein complex with the same RNA [103]. This may imply that competition between APOBEC3s and NC on viral RNA may not occur to any significant extent in mature virions. Another hypothesis to explain the observed inefficient viral particle production in the presence of mA3 in a deaminase-independent manner is that mA3 binding to MuLV gRNA in infected cells may change the metabolism of gRNA to limit the gRNA copy number available for the assembly process.

Over 100 types of RNA modification on both noncoding and messenger RNAs have been identified so far, and knowledge of their physiological roles has been accumulating [186]. Of these modifications, abundant N^6^-methyladenosine (m^6^A) modifications were identified in cellular mRNAs [187]. The development of a methodology utilizing next-generation sequencing after m^6^A-modified RNA isolation (m^6^A-seq) has enabled the location of the modification on RNAs to be identified [188]. The m^6^A modification on cellular transcripts influences a wide range of RNA biology including RNA splicing, RNA nuclear export, translation, and RNA degradation in the processing-body (P-body), which is a cytoplasmic compartment involving mRNA stability and storage [189,190,191,192,193,194,195,196,197,198,199,200]. This modification was also discovered on viral RNAs including influenza virus and adenovirus over 40 years ago [201,202,203,204,205,206]. It has become apparent that m^6^A modification on viral RNAs affects the viral life cycle [207,208,209,210,211,212,213,214,215,216,217]. For example, the m^6^A modification of MuLV gRNAs promotes viral gene expression and replication [214]. Another type of RNA modification, 5-methylcytosine (m^5^C), is also detected on cellular and viral RNAs, and has critical roles in RNA metabolism [214,218,219,220,221]. For example, the m^5^C modification on HIV-1 gRNA affects the alternative splicing of viral RNAs and the efficiency of translation [220]. Additionally, the m^5^C modification also accumulates on MuLV gRNA, and this modification appears to be required for nuclear export of the gRNA [214,221]. The m^5^C residue on cellular mRNA is recognized by the YBX1 protein, and the mRNA is stabilized [222,223]. Courtney et al. reported several distinct types of RNA modification on MuLV gRNA in addition to m^6^A and m^5^C [214]. These modifications include 2′ O-methylated ribonucleotides (Am, Gm, and Cm), 1-methylguanosine (m^1^G), and 7-methylguanosine (m^7^G). It was recently reported that cytidines of HIV-1 RNAs are acetylated by NAT10, and this modification results in stabilization of viral RNAs [224]. However, it remains largely unknown as to whether APOBEC3s impact RNA modification patterns on viral gRNA via its binding to gRNA (Figure 2). Writer, eraser, and reader proteins in RNA modification, are molecules that add, delete, and recognize modification, respectively [225]. Among them, writer proteins are predominantly localized in the nucleus, while some APOBEC3s including mA3 are excluded from the nucleus. Thus, it remains unclear whether all APOBEC3s can interact with writer proteins. However, RNA modification and its function can be executed not only in the nucleus but also in the cytoplasm [217]. It might be possible that mA3 alters the pattern of RNA epitranscriptome on MuLV gRNA in a deaminase-independent manner via its binding to gRNA or the regulatory factors in the cytoplasm, and changes gRNA metabolism (for example, RNA stability or cellular localization), leading to less efficient assembly and reduced virion production. As mentioned above, MV gRNA was reduced in the presence of APOBEC3s, although its mechanism remains unknown [155]. Thus, it would be interesting to investigate MuLV and other viral gRNA epitranscriptomes in the presence of physiological expression levels of deaminase-active or deaminase-dead mA3, and to evaluate its functional roles in future studies.

## 8. Conclusions

All human and mouse APOBEC3s interact with DNA and RNA. The enzymatic activity exerted on mammalian genomes and viral single-stranded DNA has been well characterized. In contrast, the importance of a deaminase-independent function on cellular and viral RNAs is apparent, but its role remains obscure. Further studies deciphering this role are required for an improved understanding of APOBEC3 function. These analyses may enable the identification of a virus replication process that can be targeted by novel antiretroviral drugs.

## Figures and Tables

**Figure 1 microorganisms-08-01976-f001:**
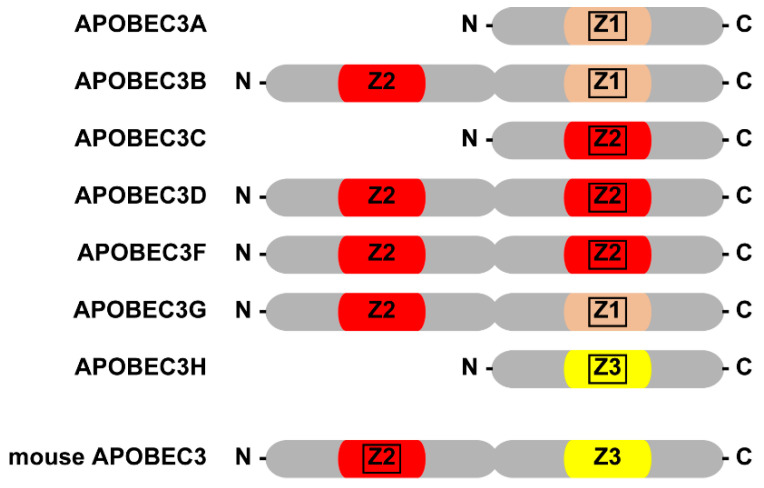
Z-domains of human APOBEC3s and mouse APOBEC3. Z1, Z2, and Z3 are shown by different colors and enzymatic active ones are marked by square box.

**Figure 2 microorganisms-08-01976-f002:**
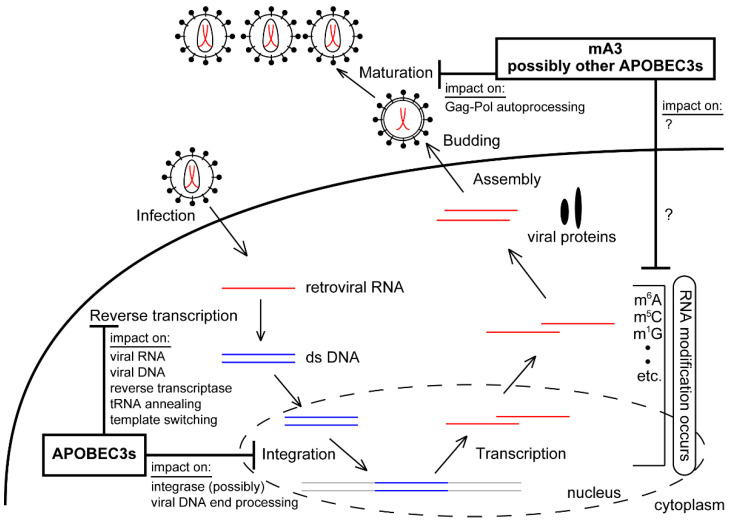
Deaminase-independent antiretroviral function of APOBEC3s. APOBEC3s inhibit the steps of the retroviral life cycle such as reverse transcription, integration, maturation (Gag-Pol autoprocessing), and possibly viral RNA modification in a deaminase-independent manner.

**Figure 3 microorganisms-08-01976-f003:**
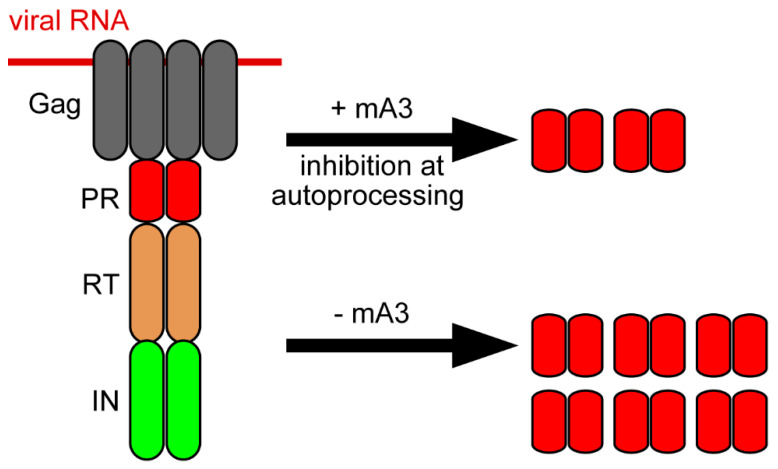
Effect of mA3 on Pr180gag-pol autoprocessing. In the presence of physiological amount of mA3, MuLV Pr180gag-pol autoprocessing is dysregulated, leading to a reduction in mature PR, and possibly RT and IN as well, in virions.

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
