# Peer review of "Deaminase-Independent Mode of Antiretroviral Action in Human and Mouse APOBEC3 Proteins"

_microorganisms, 2020, doi:10.3390/microorganisms8121976_

Round 1
Reviewer 1 Report
The revised version has much improved.
Reviewer 2 Report
The authors have addressed all my comments and suggestions and presented a much-improved manuscript. Therefore, in my opinion, it is suitable for publication.
This manuscript is a resubmission of an earlier submission. The following is a list of the peer review reports and author responses from that submission.
Round 1
Reviewer 1 Report
In this review, Hakata and Miyazawa. provided an overview of deaminase-dependent and -independent roles of APOBEC3 (A3) cytidine DNA deaminases, with a specific focus on mouse A3 (mA3) in restriction of MLV and other retroviruses. The review is very interesting, especially since the authors did not only summarize the literature on A3s but also discussed possible, undetermined, speculative deaminase-independent role (s) of mA3 on MLV inhibition such as viral epitranscriptomics. In general, the manuscript is well organized, informative, covered broader areas of the A3/retrovirus world. I would suggest the following minor points though.
Having a figure or two that self-explain the theme of this review would be a welcome addition for the reader (depicting retrovirus lifecycle vs A3 inhibition points vs catalytic-dependent or not…etc and probably different A3s with their domains single/double vs catalytic region).
The English language can be improved to make this article even better and easily understandable. Proof-reading by a native speaker or professional agency is strongly recommended.
Specific comments
- Although it covers most of the A3 functions, still, if the focus of this manuscript is going to be majorly on mA3, this can be added to the title, in my opinion. The term “APOBEC3” (as a singular) needs to specify which A3 it refers to. In the title and the abstract, the term was used to represent everything. This is mainly because different A3s have different functions. An example: “Since the discovery of APOBEC3 as a restriction factor for HIV-1, the deaminase-independent mode of action has been observed” the may not tell the readers which A3 was discovered as a restriction factor of HIV. And please use plural forms when (APOBEC3s or A3s) when giving a general statement about more than one “particular” APOBEC3, throughout this review text.
- Line 46: please specify the A3s that are correlated with cancers
- Line 50: together with REF 14, 15, please add REF 109
- Please mention the species-specificity of A3s and the discovery/function of Vif and CBF beta briefly in the earlier section
- In APOBEC4 paragraph: the recent study investigating the role of A4 on HIV-1/promoter expression (10.1371/journal.pone.0155422) can be added
- Line 130: A3G/A3C activity on the foamy virus can be included (10.1128/JVI.79.14.8724-8731.2005 and 10.1128/JVI.03385-12), as well as inhibition of EBV by A3B (10.1038/s41564-018-0284-6) can be discussed in the relevant passage
- Line 146: please add that A3C in the list of proteins that can be degraded by HIV-1 Vif
- Line 172: A3C is another single domain enzyme here to be included as well (some A3C literature: 10.1074/jbc.M408802200, 10.1038/nsmb.2378, 10.1016/j.jmb.2017.03.015, 10.1016/j.jmb.2020.10.014)
- Lines 183-4: Recent articles from Harris/Ross lab (10.1084/jem.20200261) and 10.1371/journal.pgen.1008545 need to be added and discussed in the argument
- Deaminase-independent function section: A3G and RT interaction studied in this article 10.1038/s41564-017-0063-9 and an important L1 inhibition by A3C 10.1093/nar/gkt898 can be discussed
- Table: Please change the word to, “Mode of inhibition” instead of “Inhibits”
- The paragraph on page 7 can be split, too big
- Line 347-348: please check again if the study involved mA3 as well
- Section 8, from line 353, appropriately titled to indicate mA3?
- A new epitranscriptomic modification (10.1016/j.chom.2020.05.011) reporting cytidine (acetylation by NAT10) is a good addition with respect to HIV-1 inhibition. Also, for the sake of discussion, mention that the RNA modification “writer” enzymes such as METTLs are predominantly nuclear-localized, whereas not all APOBECs are in the nucleus.
Reviewer 2 Report
The review by Hakata and Miyazawa describes mechanisms of A3 proteins not involving cytidine deamination against viruses. Overall, this is a good review that will find its audience. The review would be more accessible, if each chapter is complemented by figure illustrating the main topics.
Other issues need to be addressed:
- ”Humans have at least seven apobec3 genes” is not correct: Protein designations are the same as the gene symbols except that they are not italicized, thus it is “APOBEC3 genes”
- Typo L140: APOEBC3G
- Table 1 in its current form is not nice. It should be enhanced, include the viral system and the detailed finding, e.g., which step in reverse transcription is blocked…
- An important reference need to be included and discussed: JOURNAL OF VIROLOGY, Sept. 2011, p. 9314–9326 Vol. 85, No. 18; APOBEC3G Complexes Decrease Human Immunodeficiency Virus Type 1 Production. Kenneth L. Martin, Megan Johnson, and Richard T. D’Aquila.
- The following reference need to be included and discussed:. J Virol ,. 2020 Aug 31;94(18): Murine Leukemia Virus P50 Protein Counteracts APOBEC3 by Blocking Its Packaging. Wenming Zhao , Charbel Akkawi , Marylène Mougel , Susan R Ross. -->How can mA3 be either inhibit GAG processing and be counteracted a new GAG protein containing protein p50 at the same time? Here the authors have to develop a model to include both observations.